# Impact of HLA-B27 and Disease Status on the Gut Microbiome of the Offspring of Ankylosing Spondylitis Patients

**DOI:** 10.3390/children9040569

**Published:** 2022-04-16

**Authors:** Matthew L. Stoll, Kimberly DeQuattro, Zhixiu Li, Henna Sawhney, Pamela F. Weiss, Peter A. Nigrovic, Tracey B. Wright, Kenneth Schikler, Barbara Edelheit, Casey D. Morrow, John D. Reveille, Matthew A. Brown, Lianne S. Gensler

**Affiliations:** 1Department of Pediatrics, University of Alabama at Birmingham (UAB), Birmingham, AL 35233, USA; 2Department of Medicine, Division of Rheumatology, University of Pennsylvania, Philadelphia, PA 19104, USA; kimberly.dequattro@pennmedicine.upenn.edu; 3Centre for Genomics and Personalized Health, Queensland University of Technology (QUT), Brisbane, QLD 4000, Australia; zhixiu.li@qut.edu.au; 4Faculty of Health, School of Biomedical Sciences, Queensland University of Technology (QUT), Brisbane, QLD 4000, Australia; 5Division of Global Migration and Quarantine, Center for Disease Control, Washington, DC 30329, USA; henna.sawhney@gmail.com; 6Department of Pediatrics and Epidemiology, Perelman School of Medicine, University of Pennsylvania, Philadelphia, PA 19104, USA; weisspa@chop.edu; 7Division of Rheumatology, Children’s Hospital of Philadelphia, Philadelphia, PA 19104, USA; 8Division of Rheumatology, Inflammation and Immunity, Brigham and Women’s Hospital, Boston, MA 02115, USA; peter.nigrovic@childrens.harvard.edu; 9Division of Immunology, Boston Children’s Hospital, Boston, MA 02115, USA; 10Department of Pediatrics, University of Texas at Southwestern, Dallas, TX 75390, USA; tracey.wright@utsouthwestern.edu; 11Department of Pediatrics, University of Louisville, Louisville, KY 40292, USA; kenneth.schikler@louisville.edu; 12Department of Pediatrics, Connecticut Children’s Medical Center, Hartford, CT 06106, USA; bedelhe@connecticutchildrens.org; 13Department of Cell, Developmental and Integrative Biology, University of Alabama at Birmingham, Birmingham, AL 35294, USA; caseym@uab.edu; 14Department of Internal Medicine, University of Texas at Houston, Houston, TX 77030, USA; john.d.reveille@uth.thc.edu; 15Genomics England, London EC1M 6BQ, UK; matt.brown@kcl.ac.uk; 16Guy’s and St Thomas’ NIHR Biomedical Research Centre, King’s College, London SE1 7EH, UK; 17Department of Medicine, Division of Rheumatology, University of California at San Francisco, San Francisco, CA 94143, USA; lianne.gensler@ucsf.edu

**Keywords:** HLA-B27, microbiota, spondyloarthritis

## Abstract

Multiple studies have shown the microbiota to be abnormal in patients with spondyloarthritis (SpA). The purpose of this study was to explore the genetic contributions of these microbiota abnormalities. We analyzed the impact of HLA-B27 on the microbiota of children at risk for SpA and compared the microbiota of HLA-B27+ pediatric offspring of ankylosing spondylitis (AS) patients with that of HLA-B27+ children with SpA. Human DNA was obtained from the offspring for determination of HLA-B27 status and polygenic risk score (PRS). Fecal specimens were collected from both groups for sequencing of the V4 region of the 16S ribosomal RNA gene. Among the offspring of AS patients, there was slight clustering by HLA-B27 status. After adjusting for multiple comparisons, five operational taxonomic units (OTUs) representing three unique taxa distinguished the HLA-B27+ from negative children: *Blautia* and *Coprococcus* were lower in the HLA-B27+ offspring, while *Faecalibacterium prausnitzii* was higher. HLA-B27+ offspring without arthritis were compared to children with treatment-naïve HLA-B27+ SpA. After adjustments, clustering by diagnosis was present. A total of 21 OTUs were significantly associated with diagnosis state, including *Bacteroides* (higher in SpA patients) and *F. prausnitzii* (higher in controls). Thus, our data confirmed associations with *B. fragilis* and *F. prausnitzii* with juvenile SpA, and also suggest that the mechanism by which HLA-B27 is associated with SpA may not involve alterations of the microbiota.

## 1. Introduction

Spondyloarthritis (SpA) has a prevalence of about 1–2% of the adult population [1] and can result in significant disability and reduced quality of life. The cause appears to be multifactorial, with both genetic and environmental contributing factors. The best-characterized genetic risk factor is the HLA-B27 allele, which is present in 80–95% of white ankylosing spondylitis (AS) patients [2] and 38–68% of juvenile SpA patients [3] compared to about 7.5% of the U.S. white population [4]. However, carriage of HLA-B27 is insufficient by itself to cause AS, which develops in <5% of people with this allele [5]. While other genetic factors are clearly involved [6], there has been substantial interest in environmental factors, particularly the human intestinal microbiota. Specifically, multiple studies have demonstrated alterations in the contents and function of the intestinal microbiota in patients with pediatric and adult SpA [7,8,9,10,11,12].

It has been hypothesized that HLA-B27 itself mediates disease, at least in part, by acting upon the microbiota [13]. To test this hypothesis, we recruited a cohort of pediatric offspring of HLA-B27+ patients with AS. We compared the microbiota of the HLA-B27+ and HLA-B27- offspring and subsequently compared the microbiota of the HLA-B27+ offspring to that of treatment-naïve children with SpA. We hypothesized that the HLA-B27+ offspring, particularly those with a strong genetic burden for AS, would demonstrate an arthritogenic microbiota, and that this microbiota would be recapitulated in the children with SpA.

## 2. Materials and Methods

### 2.1. Overview

This was a cross-sectional study that first compared the contents of the fecal microbiota among HLA-B27+ versus HLA-B27- offspring of patients with AS, and subsequently compared the microbiota of treatment-naïve children with HLA-B27+ SpA with that of the healthy HLA-B27+ offspring.

### 2.2. Subjects

Controls were a sample of children of patients with AS identified by a single rheumatologist (LG) as per the modified New York criteria [14]. Children aged 5–17 years were recruited into the study and examined by a Rheumatologist (KD) for evidence of SpA. Subjects identified as having SpA were included in the SpA group.

Juvenile SpA subjects were either children with enthesitis-related arthritis as per the International League of Associations for Rheumatology criteria [15] or who met the Assessment of Spondyloarthritis International Society criteria for axial SpA [16] and were HLA-B27 positive. They were a convenience sample enrolled from six sites across the country who routinely see juvenile SpA patients (University of Alabama at Birmingham (MLS), Children’s Hospital of Philadelphia (PFW), Boston Children’s Hospital (PAN), University of Texas at Southwestern Medical Center (TW), University of Louisville (KS), and Connecticut Children’s Medical Center (BE)), in addition to one subject diagnosed with SpA as part of the assessment of the offspring of AS patients. Subjects with prior or current exposure to disease-modifying antirheumatic drugs were excluded, as were those with exposure to antibiotics within three months prior to the sample collection. All but three of the subjects were included in previous publications [7,17].

### 2.3. Processing of Fecal Samples

This was conducted as previously reported [17]. Briefly, subjects collected the samples at home and immediately placed them in a container filled with Cary-Blair media [18] and shipped them overnight to the Microbiota Core at UAB. Microbial DNA was isolated with the Zymo MiniPrep kit (cat # D6010) as per the manufacturer’s instructions.

### 2.4. Sequencing and Analysis of 16S Ribosomal DNA from the Fecal Specimens

The purified DNA (~100 ng) underwent PCR amplification using primers designed to amplify the conserved region flanking the V4 region from the 16S ribosomal RNA (rRNA) gene, as described previously [7,17]. Resulting PCR fragments were run on the Illumina MiSeq (San Diego, CA, USA) at a concentration of 12 pM; read lengths were approximately 250-bp paired-end reads. Following quality control steps, clustering was performed with the Quantitative Insight into Microbial Ecology (QIIME) platform [19], using uclust [20] as implemented by the pick_open_reference_otus.py script in QIIME. The resulting biom table and phylogenetic tree files were imported into Phyloseq [21] for further analyses, namely removal of rare operational taxonomic units (OTUs) and transformation. The filtered dataset was used for assessment of beta diversity (dissimilarity between groups) and to identify predictive microbial and clinical predictors of the structure of the microbiota. Alpha diversity richness (distinct OTUs within a sample) was assessed with the Chao1 test, while evenness (distribution of OTUs within a sample) was assessed with the Shannon and inverse Simpson measures [22]. To evaluate whether the samples clustered based upon either HLA-B27 status (for comparison of the offspring) or disease status (for comparison of the HLA-B27+ offspring to diagnosed SpA patients), the permutational analysis of variance (PERMANOVA) test was run against the distance matrix generated from the Bray Curtis measure of dissimilarity [23]. The PERMANOVA test partitions a distance matrix among sources of variation (e.g., presence versus absence of disease) in order to quantitate the strength and significance a variable has in determining the variation of distances [24]. Pairwise comparisons between the groups was performed with the DeSeq2 test [25]. Although initially designed for analysis of RNAseq data, DeSeq2 is well-suited to the analysis of microbiota data, which is also characterized by large numbers of comparators, many of which are present at near-zero levels, thus imposing often impossibly high barriers for corrected statistical significance. DeSeq2 normalizes for total read counts; filters out very low abundant features, which have a very low a priori likelihood of being informative; flags outliers based upon the Cook’s distance for either removal or imputation depending on the group sample size; and calculates the log_2_ fold change between the two groups, while correcting for multiple comparisons with the Benjamini–Hochberg false discovery rate (FDR) test [26] with a corrected significance threshold of 0.05.

As a complementary tool to tease out which taxa were most essential in distinguishing the two groups of subjects, we used the random forest algorithm [27], which is a form of decision tree analysis suitable for the analysis of “outcome” (here, HLA-B27 positive versus negative) of microbiota data. We incorporated both OTU-level information as well as basic demographic features in the model.

Differences in total read counts and in alpha diversity were evaluated using the Student’s t-test, with an alpha of 0.05. Differences in the PRS by HLA-B27 status were assessed with the nonparametric Kruskal–Wallis test.

### 2.5. Genotyping and PRS Calculation

Calculation of the PRS was performed as recently described by our group [28], using the PRS developed and validated for white European ancestry subjects. Briefly, samples were genotyped using the Illumina Core-Exome SNP genotyping microarray per manufacturer’s instructions, with genotypes called using the Genome Studio V.2.0 software available on Illumina.com. *HLA-B27* imputation was performed with SNP2HLA [29] against the Haplotype Reference Consortium panel [30].

## 3. Results

### 3.1. Offspring of Ankylosing Spondylitis Patients

A total of 56 offspring of AS patients met inclusion criteria for the current study and submitted a fecal specimen. The PRS was obtained on 55 of the offspring, although we were still able to impute HLA-B27 status on the subject without a PRS. One of the subjects was diagnosed with SpA at the research exam on the basis of arthritis, enthesitis, and clinical sacroiliitis. That subject was found to be HLA-B27+ by genotyping, subsequently confirmed through clinical testing, and was, therefore, included in the SpA group. Of the 56 offspring, 32 (57%) were found to be HLA-B27 positive. A description of the probands with AS is included in Table 1, while a comparison of the clinical and demographic features of the HLA-B27 positive and negative offspring is shown in Table 2.

There were no differences in alpha diversity of raw read counts between HLA-B27+ and HLA-B27- offspring (Figure 1). Ordination analysis demonstrated no visually evident separation between the two groups of subjects (Figure 2), although the PERMANOVA test did demonstrate slight clustering (F = 1.8, *p* = 0.022). There likewise was association between the PRS and the structure of the microbiota (F = 1.8, *p* = 0.017). However, a major driver of the PRS is HLA-B27 itself; the PRS of the HLA-B27- offspring was −0.25 (IQR −0.30, −0.21), while that of HLA-B27+ offspring was 0.22 (IRQ 0.1, 0.41, *p* < 0.001). There was no association between the PRS and the structure of the microbiota when the analyses were performed separately on HLA-B27+ and HLA-B27- offspring. For HLA-B27+ subjects, the F-score was 1.5 (*p* = 0.081) and, for their HLA-B27- counterparts, the F score was 1.1 (*p* = 0.345).

Next, we used the DeSeq2 program [25] to perform pairwise comparisons between the groups. Not unexpectedly in light of the overall similarity between the structure of the microbiota between the two groups (Figure 2), only five OTUs differentiated the two sets of subjects (Table 3, Appendix A) Of these, *F. prausnitzii* had higher abundance in the HLA-B27+ subjects, while *Coprococcus* and three OTUs from the *Blautia* genus were all higher in the HLA-B27- subjects.

As a complementary tool to tease out which taxa were most essential in distinguishing the two groups of subjects, we used the random forest algorithm [27], which is a form of decision tree analysis suitable for the analysis of “outcome” (here, HLA-B27+ versus negative) of microbiota data. We incorporated both OTU-level information as well as basic demographic features in the model. The 10 most distinguishing factors included four distinct OTUs belonging to *F. prausnitzii* (Figure 3). The reason that multiple OTUs matched to the same organism is that there are multiple entries in the database for bacteria representing subspecies or strains that are more than 3% different from one another, as indicated by the unique identifiers shown on the *Y*-axis. Thus, multiple unique strains of *F. prausnitzii* were associated with HLA-B27 status, underscoring its association.

### 3.2. Comparison of HLA-B27+ SpA Subjects and Offspring

A total of 32 offspring were identified, including the one found to have SpA at the initial assessment. A total of 23 children diagnosed with HLA-B27+ SpA and naïve to immunomodulatory therapy were identified elsewhere, for a total of 24 HLA-B27+ SpA subjects and 31 HLA-B27+ offspring without arthritis. As depicted in Table 4, the children with SpA were more likely to be male and were slightly older than the HLA-B27+ offspring; in addition, 29% of the SpA patients versus none of the controls reported use of NSAIDs at the time of sample collection.

Alpha diversity analyses demonstrated that the SpA patients had decreased richness (Chao1 test) and evenness (Shannon test) as compared to the controls, although a second test of evenness (inverse Simpson) did not demonstrate any differences between the two groups (Figure 4). Ordination analysis (Figure 5) demonstrated partial separation of the two sets of subjects, confirmed by the PERMANOVA test (F = 2.5; *p* = 0.001), which remained significant following adjustment for NSAID use, age, and sex (F = 1.6, *p* = 0.003). DeSeq2 analysis demonstrated 21 OTUs that distinguished the patients from controls (Table 5, Appendix A). Consistent with our previous work [7,17], *Bacteroides* was higher in the SpA patients, while *F. prausnitzii* was lower in SpA patients. Of *Blautia* and *Coprococcus*, both of which were higher in the HLA-B27- versus HLA-B27+ siblings, one of them (*Blautia*) was higher in the SpA patients versus the HLA-B27+ siblings, while the other (*Coprococcus*) was lower in the SpA patients. As discussed above with respect to the random forest plot, several organisms appear more than once, due to multiple entries representing species or strains that cannot be distinguished at the 16S level.

## 4. Discussion

This is the first study to compare the contents of the fecal microbiota of HLA-B27+ versus HLA-B27- offspring of AS patients and is also the first to compare that of HLA-B27+ pediatric subjects with and without SpA. Several important messages emerged from this work.

First, while the HLA-B27 allele may influence the contents of the fecal microbiota, this work does not support the hypothesis that the impact of this allele on the microbiota is the mechanism whereby it influences the risk of SpA. Prior studies evaluating the impact of HLA-B27 on the microbiota have yielded contradictory findings. Specifically, Breban and colleagues compared the fecal microbiota of HLA-B27 positive to negative siblings of SpA patients [10]; similar to the present study, the PERMANOVA test demonstrated no differences between the two groups, while pairwise testing demonstrated differences in rare bacteria. A study of AS patients likewise revealed minimal differences between HLA-B27 positive and negative subjects [31], although, in this study, only a subset of the microbiome was profiled, and most patients were receiving therapies. In contrast, Asquith and colleagues [32] compared the contents of the microbiota of HLA-B27+ versus negative healthy adults undergoing ileocolonoscopy, showing substantial and consistent differences at multiple habitats within the gut, including decreased *Blautia* among the HLA-B27+ subjects. In the current study, a key organism distinguishing HLA-B27+ from negative subjects was *F. prausnitzii*, which, despite being depleted in juvenile SpA patients ([7] and present study) and generally being considered a regulatory organism [33], was present in increased abundance in the HLA-B27 positive cohort. Along those lines, this study does not provide any evidence that the overall genetic milieu associated with SpA, independent of HLA-B27, impacts the microbiota, although the study’s power to assess this is modest. Further, the PRS applied is optimized for use in white European ancestry subjects, whereas the participants of this study were of diverse ancestry.

Second, our study reconfirms findings of an altered microbiota in patients with SpA. This has been demonstrated in multiple prior studies of patients with AS or undifferentiated SpA [7,8,9,11,12,17,34]. Additionally, several of the taxa identified herein have previously been linked to this condition. Specifically, Aggarwal and our own group have previously demonstrated increased *Bacteroides* in children with SpA [7,11,17], and our work as well as that of Zhou [35] have shown decreased abundance of *F. prausnitzii*. In contrast to the present work, *Blautia* has been reported to be increased in AS [36], while there are contradictory data with respect to the direction of change of *Coprococcus* [8,12].

Finally, this study appears to underscore the importance of age or disease state in the nature of the microbial abnormalities. We have previously demonstrated that the fecal abundance of *F. prausnitzii* is decreased in children with juvenile SpA [7], while increased abundance was observed in the AS probands of the current study as compared to healthy HLA-B27+ adults. Additionally, this study demonstrated increased abundance of *Bacteroides* in general and *B. ovatus* specifically in children with SpA. This organism has frequently been reported as abundant in pediatric arthritis, including) but not limited to SpA [7,11,37,38], while the opposite is observed in adults with SpA [17,31,34,39,40]. Increased abundance of bacteria that may have a regulatory impact on immune function may reflect altered mucosal immune development, which could, in turn, predispose to autoimmunity [41].

This study has limitations, including the potential for differences due to the age differences between the HLA-B27+ offspring and the SpA patients. We do not suspect this difference to be problematic, as the microbiota appears to stabilize after age 3 [42]. Age did not impact clustering in either group, and the differences in the contents of the microbiota remained significant after adjustments were made for age. It is also possible that the observed differences between the HLA-B27+ patients and the HLA-B27+ offspring represented geographic variation. Our previous [17] work did not reveal significant geographic variation of the intestinal microbiota among children with SpA, and, among inhabitants of developed nations, the microbiota may not vary widely even across countries [43], although further studies in which the patients and controls are drawn from the same geographic locality would be optimal. Our study also has a modest sample size.

A novel aspect of this work is that of comparing both HLA-B27+ and HLA-B27- offspring who are well-balanced for age, sex, and geographic factors, and also comparing HLA-B27+ SpA patients to HLA-B27+ controls. Prior studies of SpA patients [17,31,34,39] had compared SpA patients to healthy controls, the majority of whom are HLA-B27-. Ours is the first pediatric study limited to individuals with this risk allele, with the results supporting previous pediatric studies. Future studies of SpA patients should take into account the risk allele, and additional studies are required to validate the findings reported herein.

## Figures and Tables

**Figure 1 children-09-00569-f001:**
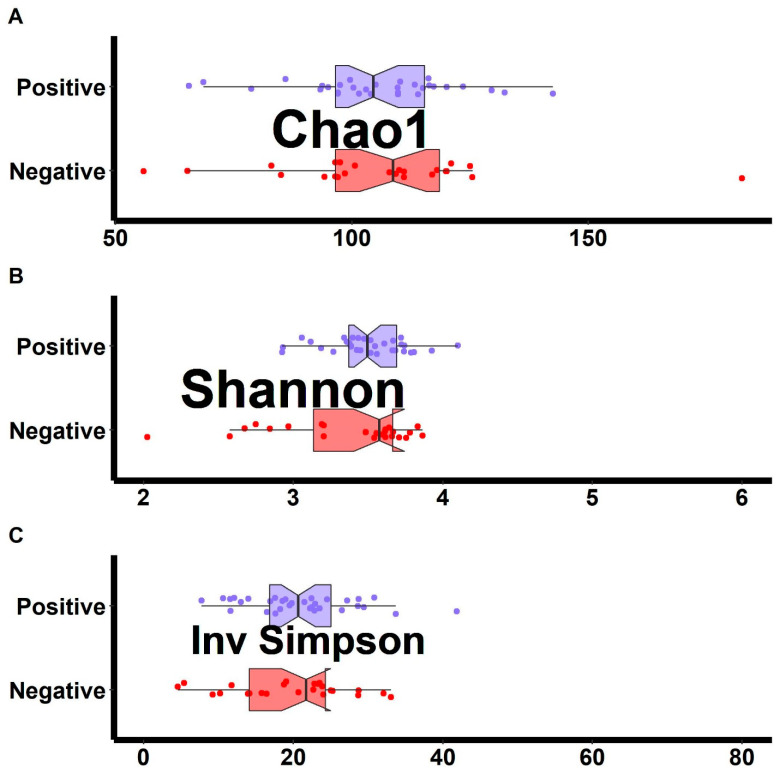
Alpha diversity measures comparing HLA-B27+ to HLA-B27- offspring of AS patients. The Chao1 measure of richness (**A**) along with the Shannon (**B**) and inverse Simpson measures of evenness (**C**) are shown.

**Figure 2 children-09-00569-f002:**
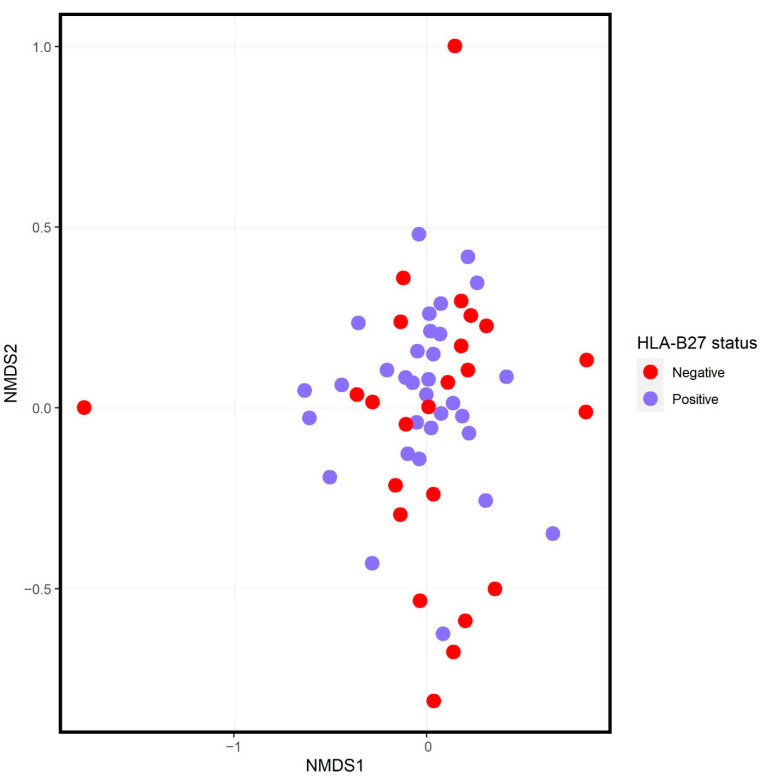
Ordination analysis of HLA-B27 positive (blue) versus negative (red) subjects. NMDS = nonmetric multidimensional scaling.

**Figure 3 children-09-00569-f003:**
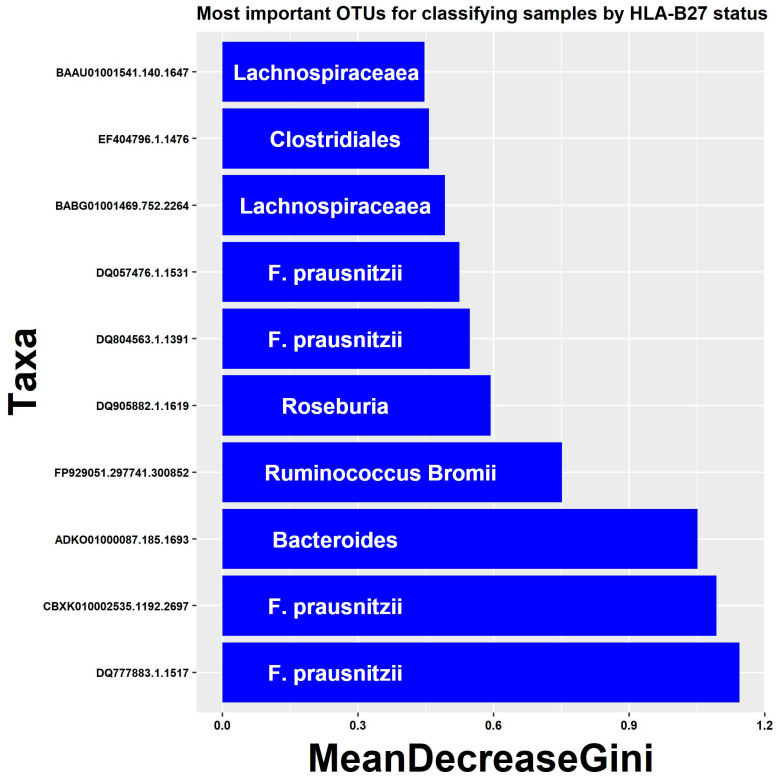
Results of random forest algorithm to identify features associated with HLA-B27 status. The top 10 most important features based upon the MeanDecreaseGini coefficient are shown.

**Figure 4 children-09-00569-f004:**
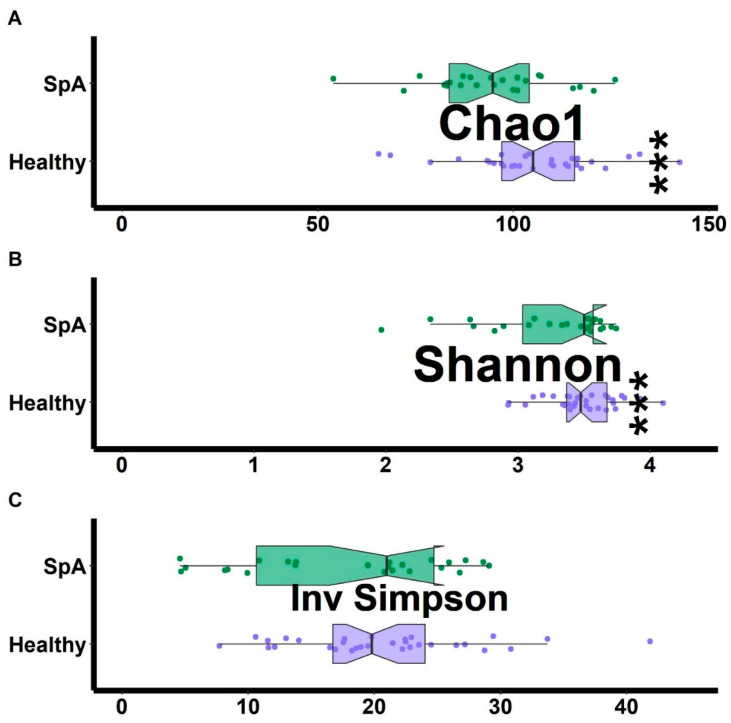
Alpha diversity measures comparing HLA-B27+ offspring of AS patients to HLA-B27+ treatment-naïve children with SpA. The Chao1 measure of richness (**A**) along with the Shannon (**B**) and inverse Simpson measures of evenness (**C**) are shown. Asterisks indicate statistically significant difference (*p* < 0.05).

**Figure 5 children-09-00569-f005:**
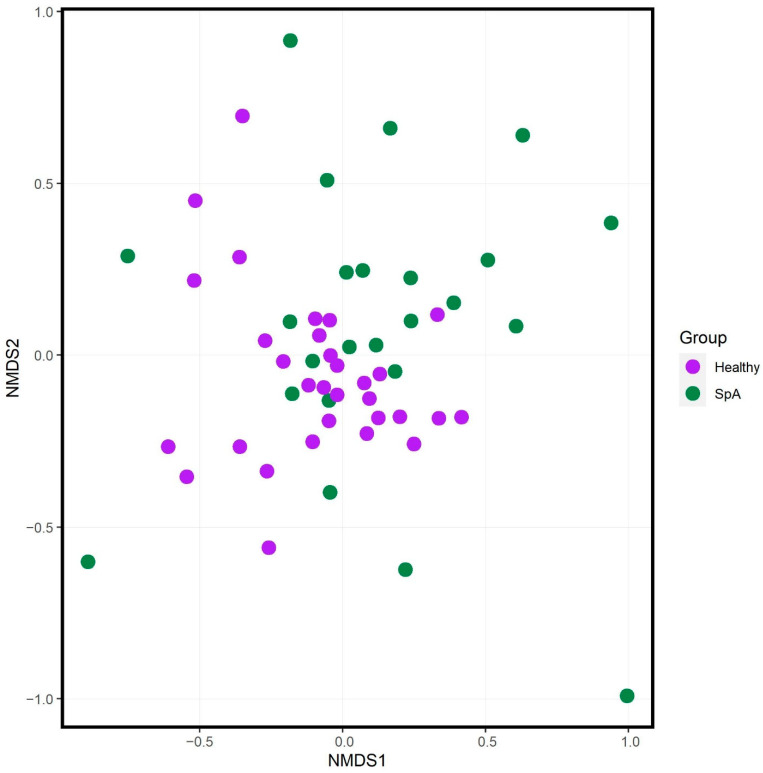
Ordination analysis of HLA-B27 positive healthy offspring (purple) versus HLA-B27+ SpA subjects (green). NMDS = nonmetric multidimensional scaling.

**Table 1 children-09-00569-t001:** Clinical features of ankylosing spondylitis probands. Abbreviations: ASDAS-CRP, ankylosing spondylitis disease activity score—C reactive protein; BASDAI, Bath ankylosing spondylitis disease activity index; BASFI, Bath ankylosing spondylitis functional index; NSAIDs, nonsteroidal anti-inflammatory drugs.

Feature	Value
n	31
*Demographics*	
Male Sex	23 (74.2%)
Age (years)	45.7 ± 7.1
Race	
White	20 (64.5%)
Asian	8 (25.8%)
American Indian	1 (3.2%)
Multiracial	2 (6.5%)
Hispanic ethnicity	3 (9.7%)
Treatment	
NSAIDs	18 (58.1%)
Methotrexate	1 (3.2%)
Etanercept	5 (16.1%)
TNFi mAb	11 (35.5%)
Secukinumab	1 (3.2%)
Body mass index (mg/kg^2^ ± SD)	25.5 ± 4.0
BASDAI	2.5 ± 2.3
BASFI	1.7 ± 2.4
ASDAS-CRP	1.8 ± 1.0

**Table 2 children-09-00569-t002:** Demographic and clinical variables of the study population. Abbreviations: NSAIDs, nonsteroidal anti-inflammatory drugs.

Feature	HLA-B27 Negative	HLA-B27 Positive
n	24	32
*Demographics*		
Male Sex	12 (50%)	12 (38%)
Age (years)	10.5 ± 4.0	10.7 ± 3.7
Race		
White	11 (46%)	11 (34%)
Asian	4 (17%)	4 (12%)
American Indian	0	2 (6.2)
Multiracial	9 (38%)	15 (47%)
Hispanic ethnicity	6 [25]	5 (16%)
NSAID usage	2 (8.3%)	1 (3.1%)
Body mass index (mg/kg^2^ ± SD)	17.5 ± 3.2	18.0 ± 3.8

**Table 3 children-09-00569-t003:** Statistically significant results of the DeSeq2 output comparing HLA-B27+ to negative offspring. The full dataset is shown as Appendix A. Abbreviations: LFC, log2Fold change; LFCSE, log2Fold change standard error; padj, adjusted (corrected) *p*-value. LFC values > 0 represent OTUs higher in HLA-B27+ subjects.

Organism	BaseMean	LFC	LFCSE	padj
*Blautia*	2539	−1.91	0.40	6.13 × 10^−5^
*Coprococcus*	1146	−1.35	0.38	0.005
*Blautia obeum*	1142	−2.17	0.55	0.001
*Blautia*	743	−1.31	0.42	0.016
*Faecalibacterium prausnitzii*	611	0.72	0.26	0.034

**Table 4 children-09-00569-t004:** Demographic and clinical variables of the HLA-B27+ study population. Abbreviations: NSAIDs, nonsteroidal anti-inflammatory drugs, SpA; spondyloarthritis. The number of offspring is one less than that reported in Table 1, as the subject with SpA was included in the SpA group.

Feature	Offspring	SpA
n	31	24
*Demographics*		
Male Sex	12 (38.7%)	15 (62%)
Age (years)	10.6 ± 3.7	13.5 ± 2.9
Race		
White	11 (36%)	17 (71%)
Asian	4 (13%)	1 (4.2%)
Black	0	4 (17%)
American Indian	2 (6.5%)	0
Multiracial	14 (45%)	1 (4.2%)
Unknown	0	1 (4.2%)
Hispanic ethnicity	5 (16.1%)	0
NSAID usage	0	7 (29.2%)
Sacroiliitis	0	17 (71%)
Enthesitis	0	11 (46%)
Acute anterior uveitis	0	0
Body mass index (mg/kg^2^ ± SD)	17.8 ± 3.7	18.1 ± 3.8

**Table 5 children-09-00569-t005:** Statistically significant results of the DeSeq2 output comparing HLA-B27+ healthy offspring to HLA-B27+ SpA subjects. The full dataset is shown as Appendix A. Abbreviations: LFC, log2Fold change; LFCSE, log2Fold change standard error; padj, adjusted (corrected) *p*-value. LFC values > 0 represent OTUs higher than the SpA subjects.

Organism	BaseMean	LFC	LFCSE	padj
*Bacteroides*	3233	1.27	0.47	0.047
*Blautia*	1803	1.47	0.35	<0.001
*Bacteroides ovatus*	1132	1.12	0.40	0.040
*Faecalibacterium prausnitzii*	618	−1.19	0.38	0.018
*Escherichia coli*	277	2.59	0.78	0.011
*Parabacteroides*	274	2.92	1.03	0.040
*Bifidobacterium*	128	−2.12	0.59	0.004
*Coprococcus*	109	−1.95	0.51	0.002
*Ruminococcus*	89	−26.0	2.47	<0.001
*Ruminococcus torques*	78	2.96	0.62	<0.001
Unspecified *Lachnospiraceae*	67	−2.12	0.76	0.047
*Coprococcus*	53	−2.22	0.49	<0.001
*Bacteroides*	50	3.01	0.97	0.018
Unspecified *Clostridiales*	46	−10.8	2.44	<0.001
Unspecified *Christensenellaceae*	43	−5.96	2.01	0.043
*Bacteroides eggerthii*	27	4.49	0.89	<0.001
*Ruminococcus*	20	−24.2	2.94	<0.001
*Bacteroides plebeius*	14	−9.59	2.94	0.013
*Eubacterium biforme*	11	−24.1	2.94	<0.001
*Parabacteroides*	10	2.55	0.94	0.047
*Coprococcus eutactus*	6.6	−5.87	1.81	0.013

## Data Availability

The sequence reads were deposited into the Sequence Read Archive.

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
