# Peer review of "Impact of HLA-B27 and Disease Status on the Gut Microbiome of the Offspring of Ankylosing Spondylitis Patients"

_children, 2022, doi:10.3390/children9040569_

Round 1

Reviewer 1 Report

The authhors have analysed the potential impact of HLA-B27 on the microbiota of offspring of AS patients. Its novelty and importance stems from the comparison of fecal microbiota of HLA-B27 positive children with and without SpA, as well as comparison of fecal microbiota of HLA-B27+ vs. HLA-B27- offspring of AS patients. Its weakness is that PRS optimized for European ancestry subjects has been applied for a cohort with diverce ancestry.

The following points should be adressed before publication:

1) revision of the title;

The title sounds quite general. It could be optimized in order to highlight the specific design of the study.

2) L76-77 - specify the six sites across the cuntry;

3) Table 1 and Table 3 - replace the terms "white" and "black" race;

4) improvement of statistical analysis description;

Statistical analyses have been poorly described. For instance, PERMANOVA test have been applied but not defined in the statistics subsection; L127 - a more precise description must replace the phrase "with an alpha of 0.05".

5) Figure 2 and Figure 5 - modify captions, define NMDS1 and NMDS2;

6) revision of Figure 3;

This figure needs better presentation. Define different F. prausnitzii OTUs.

7) Table 5 shows two different results for Bacteroides, Coprococcus, Ruminococcus, Parabacteroides. What is the reason?

Reviewer 2 Report

In this analysis, the authors compared the microbiome of offsprings of AS patients.

-I think both AS group and their offsprings should be better explained. Sites of involvement of AS group, for instance, may have some implications.

-Also, characteristics of juvenile SpA group should be given in details.

-Please use juvenile SpA term instead of pediatric SpA.

Round 2

Reviewer 1 Report

The authors have satisfactorily addressed most of my concerns. The manuscript has been improved.

I recommend the following minor revisions before publication:

1) Materials and methods include subsection Statistical analysis, which has not been improved. In fact, most of the statistical analyses have been described in subsection Sequencing and analysis of 16S ribosomal DNA. The authors could either merge the two subsections or include all statistical analyses in the corresponding subsection.

The authors have clarified in the text the reason that one organism could have multiple OTUs. However, I'm convinced that Figure 3 could be improved.

Reviewer 2 Report

All my queries were addressed. 

Author Response

Thank you for the re-review.